



# Measurement Report: Atmospheric aging of combustion-derived particles: impact on stable free radical concentration and its ability to produce reactive oxygen species in aqueous media

Heather L. Runberg[1], Brian J. Majestic[1]

[1] Department of Chemistry and Biochemistry, University of Denver, Denver, CO 80208-9020, USA

*Correspondence to*: Brian J. Majestic (brian.majestic@du.edu)





**Abstract.** Environmentally persistent free radicals (EPFR) are a pollutant found on fine atmospheric particulate matter ($PM_{2.5}$), particularly on $PM_{2.5}$ formed from combustion processes. EPFR are organic radicals that can endure in the environment for days to years. Interest in the toxicity of EPFR has increased significantly in recent years, as it has been shown to have substantial ability to form reactive oxygen species (ROS), but little is known about how its characteristics change as $PM_{2.5}$ ages in the atmosphere. Here, we exposed newly produced hexane-generated soot to simulated sunlight for 24 hours. Changes to the EPFR characteristics of the particles were measured by electron paramagnetic resonance (EPR) spectroscopy. The soot was then added to water and a second exposure to light was used to measure hydroxyl radical (OH) formation from both photo-aged and dark-aged soot. There were no changes to EPFR characteristics (spin concentration, g-factor, peak width, or lineshape) due to the exposure to simulated sunlight, however the soot's ability to form OH was greatly reduced by photo-aging. Photo-aged soot resulted in an almost 60% reduction in OH formation over soot which had been aged in the dark for the same amount of time.

## 1 Introduction

Fine atmospheric particulate matter, particles smaller than 2.5 μm in diameter ($PM_{2.5}$), are a well-documented environmental problem. Because of their small size, $PM_{2.5}$ can penetrate deep into the lungs where they have been shown to cause oxidative damage via the production of reactive oxygen species (ROS) (Feng et al., 2016; Vidrio et al., 2009). $PM_{2.5}$ has been linked to many cardiopulmonary health concerns such as asthma, high blood pressure, and COPD (He et al., 2022; Chan et al., 2008), as well as diabetes (Brook et al., 2012, 2013) and cancer (Guo et al., 2020; Katanoda et al., 2011). One of the major sources of $PM_{2.5}$ is combustion, from sources such as automobile exhaust, power plants, and wildfires. Climate change has already driven an increase in the number and severity of wildfires globally (Coop et al., 2022) and there is evidence that $PM_{2.5}$ from wildfires may be more hazardous than from other sources (Aguilera et al., 2021).

$PM_{2.5}$ can stay aloft for a very long time and can travel thousands of kilometers from its point source. During this time, it is subjected to various environmental conditions such as weather, sunlight, and interaction with atmospheric gasses like $NO_X$, $SO_X$, or reactive oxygen species (ROS, e.g. OH, $O_3$, $O^{2-}$), all of which can alter the chemical composition of $PM_{2.5}$ components, resulting in aging of the particles (Liu and Chan, 2022; Donahue et al., 2009).

Additionally, $PM_{2.5}$ has been shown to contain substantial concentrations of environmentally persistent free radicals (EPFR) (Chen et al., 2018; Runberg et al., 2020), a newly identified class of pollutant. Radicals are chemical species with at least one unpaired electron and typically are very reactive, with lifetimes generally on the scale of nanoseconds. EPFR, however, persist for days, weeks, or even years in the environment making them an important environmental contaminant (Gehling and Dellinger, 2013). EPFR are generally carbon- and oxygen-centered radicals and have been shown to form from the same combustion systems that result in $PM_{2.5}$ (Wang et al., 2019). Both $PM_{2.5}$ and EPFR have been reported to induce ROS formation (Tong et al., 2018; Arangio et al., 2016; Khachatryan et al., 2011), and it is possible that the EPFR component is a significant contributor of $PM_{2.5}$'s ability to form ROS.



To date, little is known about the impacts of atmospheric aging on EPFR bound to $PM_{2.5}$, such as its character, concentrations, or its ability to mediate ROS formation. One of the key components of $PM_{2.5}$ is soot, which contains EPFR, and as such it was chosen as a surrogate for environmental PM because its age and environmental encounters could be controlled.

In this study, the EPFR characteristics of fresh hexane-derived soot is compared to soot that has been photo aged. EPFR concentrations and characteristics are measured by electron paramagnetic resonance (EPR) spectroscopy, Fourier-transform

infrared (FT-IR) spectroscopy, and gas chromatography mass spectrometry (GC-MS) before and after illumination under a solar simulator. Additionally, the ability of both the photo-aged and non-aged soot to produce OH in aqueous media is studied, and the EPFR characteristics of the soot after OH formation is also investigated.

## 2 Materials and Methods

### 2.1 Photo-aging of combustion particles

**Stage I:** Hexane derived soot was produced as previously described (Runberg et al., 2020) by using a diffusion flame with a glass funnel inverted above the flame to accumulate the soot. The soot was collected from the funnel into a disposable aluminum baking dish (Reynolds KITCHENS® mini loaf pan) as soon as it was cool enough to handle (Figure 1-I).

**Stage II:** The bulk soot was divided roughly evenly between two additional baking dishes. One dish was placed under a xenon lamp solar simulator (Oriel Sol 1A, Newport Solar Simulator equipped with an AM 1.5 Global filter) calibrated to one sun at

the average North American zenith, and the other was kept in the dark by covering it with aluminum foil. The soot was aged in this manner for 24 hours (Figure 1-II). This process was completed twice, once to produce soot for photoreactions in stage III which included an OH probe, and once for reactions executed without an OH probe.

**Stage III:** After the soot had been aged for 24 hours, either with or without light, it was combined with ultra-pure water (18 MΩ) (Figure 1-III) and stirred overnight to create a well-mixed suspension, or slurry. The final soot concentration was 45

± 0.5 ppm (m/v) for slurries that were used for OH measurements, which is in line with concentrations of $PM_{2.5}$ obtained in previous work (Leresche et al., 2021), and 118 ± 2 ppm (m/v) for slurries used for all other analyses. Slurry concentrations for non-OH analyses were higher to allow for enough soot to be collected after the photoreactions for EPFR, FT-IR, and GC-MS analysis.

Samples of the prepared slurries were poured into two Teflon reaction vessels for the aqueous reaction in Stage III. Both

reaction vessels were placed into water-cooled, jacketed glass beakers which were kept at 25℃. One was located under the solar simulator and the other was covered with aluminum foil to keep it dark. The slurries were stirred continuously for the 16-hour reaction time. For EPFR, FT-IR, and GC-MS analyses, the reactants were poured into clean baking dishes and dried overnight in an oven set to 50℃. Once the soot was dry, it was collected for further analysis. All samples were stored in a freezer when not being analyzed.




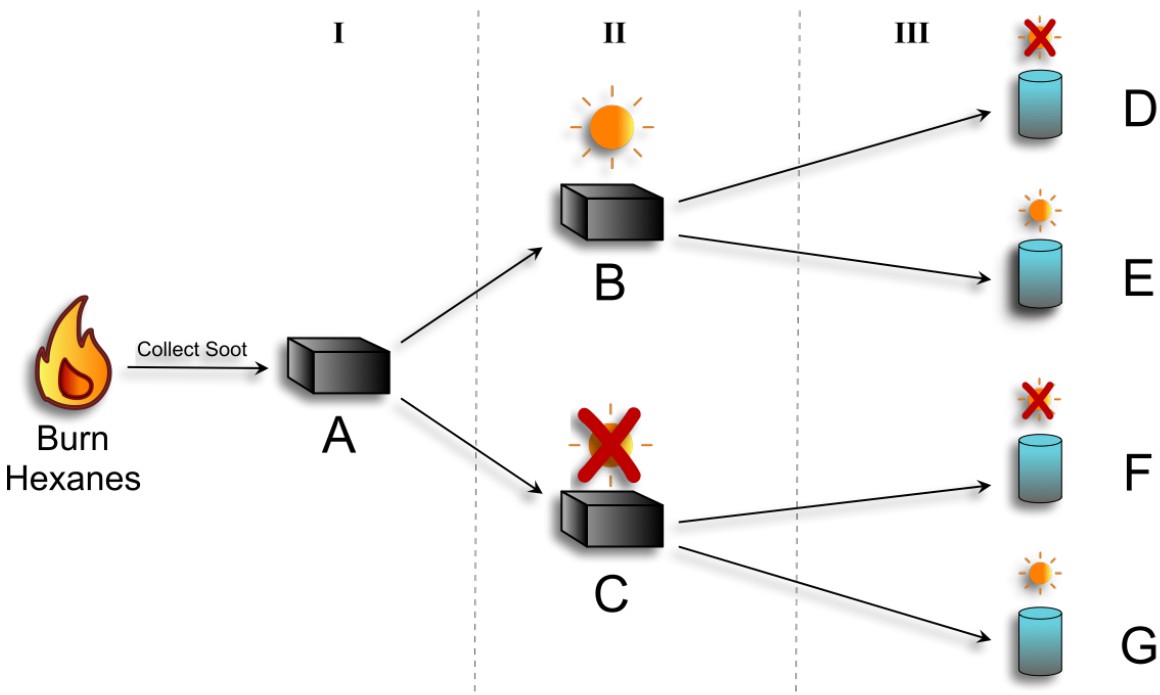

**Figure 1. Schematic of photo-aging experimental setup. Hexanes were burned and resultant soot was collected (A). That soot was divided, and half was exposed to simulated sunlight for 24 hours (B) and half was kept in the dark (C). The soot from each was then added to water to make a slurry. The two soot slurries were then divided in half, and one was exposed to simulated sunlight for 16 hours (E and G) and the other was kept in the dark (D and F). Soot characterization was done at each stage (I, II, and III), via GC-MS, FT-IR and EPR. OH concentrations were measured for stage III.**

## 2.2 EPR Analysis

Samples of soot ranging between $0.4 – 0.9 ± 0.1$ mg were collected in triplicate in 1.5 mm ID Pyrex closed-bottom capillary tubes after each stage of photo-aging for later EPR investigation; samples were capped with Critoseal and stored in a freezer until analysis. EPR analysis and EPFR characterization of soot samples were performed using previously described methods (Runberg et al., 2020). EPFR concentrations were measured using the spin-count feature available in the Bruker EMXnano Xenon software (see Table A1 for Q values and frequencies obtained). Spin counts were determined in triplicate for each sample and averaged. Lineshape evaluation, including g factors and Gaussian and Lorentzian line contributions, was performed using least squares analysis available in the EasySpin (easyspin.org) package available in MATLAB (Table A1). Triplicate soot samples were averaged together for final lineshape analysis.

## 2.3 Determination of [OH] via HPLC

Sodium benzoate (SBA) was used as an OH probe because it readily forms para-hydroxybenzoic acid (p-HBA) when exposed to OH radicals (Wu et al., 2017); p-HBA is relatively stable and can be easily quantified via high performance liquid



chromatography (HPLC). SBA (1.0 mM) was added to 45 ppm soot slurries just prior to beginning the photoreaction. Aliquots

of approximately 5 mL of soot slurry were collected before and after 16-hour photoreactions and filtered using 0.2μm, 25 mm diameter Whatman polypropylene filters. Filters were rinsed with a minimum of 15 mL ultrapure water prior to filtering. Filtrate was analyzed with no further modification via HPLC (Agilent 1100 with diode-array detector (DAD), and a C-18 Hydro-RP 250 × 4.6 mm column (Phenomenex)) using the method described previously (Runberg and Majestic, 2022). Each sample was run in triplicate and final concentrations were averaged.

95        Fresh p-HBA standards were prepared in ultrapure water on the day of analysis, with a limit of quantification of 0.11 $\pm$ 0.06 μM, as determined by dividing the standard deviation of the response by the slope of the calibration curve and multiplying by 10 (International Committee on Harmonization, 2005). Final OH concentrations were determined by dividing the measured p-HBA by the fraction of OH that forms p-HBA after 16 hours, previously determined to be 0.56 (Runberg and Majestic, 2022).

**2.4 GC-MS Analysis**

A solid-liquid organic extraction was done to obtain the organic-soluble fraction of soot for analysis via gas chromatography mass spectroscopy (GC-MS). Soot from each stage of the photo-aging process was extracted into acetonitrile (ACN) by submerging a small amount into an aliquot of ACN and allowing it to soak overnight. The ACN was then filtered using a 0.2 μm polypropylene Whatman filter to remove any remaining soot particulate before examination via an Agilent 7820A GC

system coupled to a 5977B MSD. A previously described GC-MS method was used (Runberg and Majestic, 2022) and mass spectral analysis was preformed using Agilent MassHunter Qualitative Analysis Navigator (B.08.00) and NIST MS Search (version 2.3).

**2.5 FT-IR**

Fourier-transform infrared spectroscopy (FTIR) was done for all samples to assess any bulk changes to functional groups as

the soot is aged. The system used was a D5 ATR Diamond, Thermo Scientific spectrograph. Analysis was performed by doing 30 scans per sample.

**3 Results**

**3.1 EPFR Characterization**

Figure 2 and Table A1 present the EPFR concentrations at various stages in the aging process. There was no significant

difference in EPFR concentration at any phase of the soot aging process. Concentrations of EPFR ranged from (2.5 $\pm$ 0.3) x $10^{16}$ spins mg$^{-1}$ to (3.4 $\pm$ 0.6) x $10^{16}$ spins mg$^{-1}$. These concentrations that are consistent with previous soot and biochar studies (Runberg et al., 2020; Tian et al., 2009; Sigmund et al., 2021).



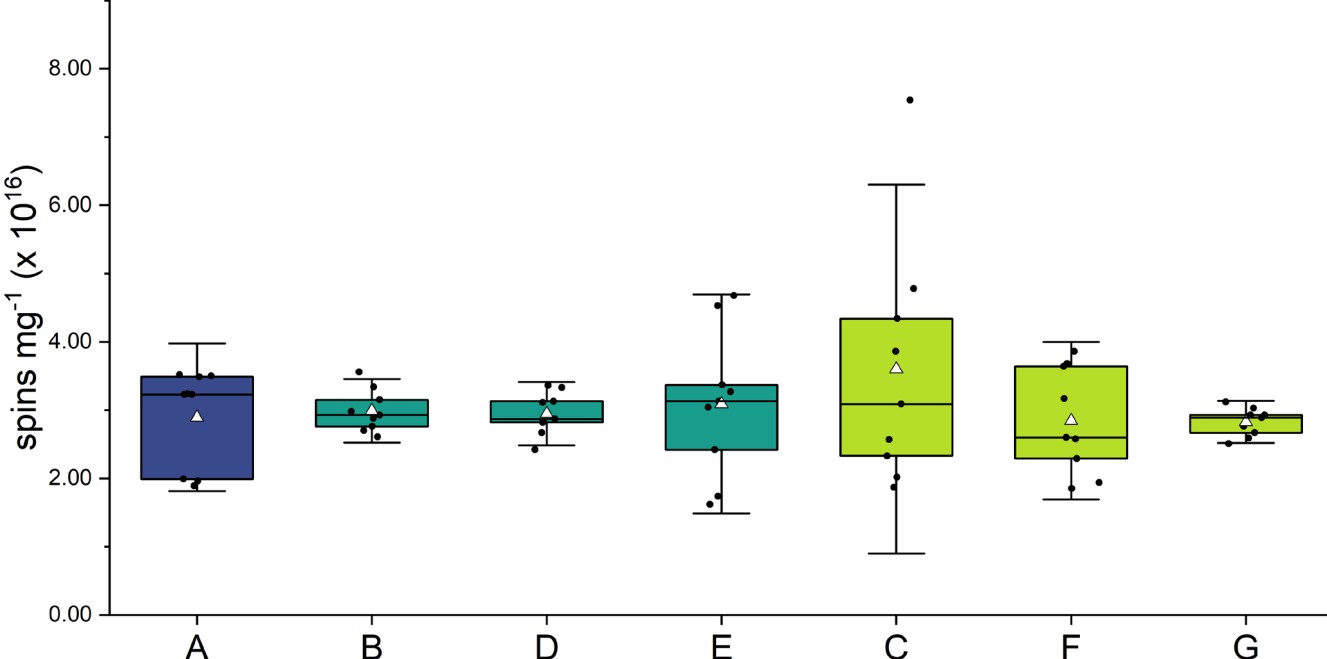

**Figure 2 EPFR concentrations (spins per mg of soot) from soot samples at each step of the aging process (A-G). Three soot samples were collected from each phase of the experiment, and spins were calculated in triplicate for each sample. Average is shown as a white triangle.**

One way to characterize an EPR signal is by determining the associated g-factor, a unitless proportionality constant determined by the ratio of the magnetic field and the applied microwave frequency. Equation 1 shows the relationship of g to the frequency

($\nu$) and the magnetic field (B), where h is Planck's constant and $\beta$ is the Bohr magnetron (Eaton et al., 2010), see Eq. (1):

$$g = \frac{h\nu}{\beta B},\tag{1}$$

Electrons are half-spin particles with two possible spin states, $+\frac{1}{2}$ and $-\frac{1}{2}$, with the difference in energy between them equal to $\Delta E$. In EPR spectroscopy, the microwave frequency ($\nu$) is usually held constant while the magnetic field (B) is modulated. When $\Delta E$ equals the applied microwave energy, resonance is achieved and absorption occurs, resulting in an EPR signal. The

g-factor at the point of resonance can be used to identify the type of radical being measured (Eaton et al., 2010; EPR-Interpretation, 2022).

In the current study, there were no changes observed to the g-factor of the soot over the course of the aging process, which was between 2.0056-2.0057 for all samples (Table A1 and Figure A1) and is consistent with the g-factor of an organic radical (EPR-Interpretation, 2022). Changes in g-factor indicate a change in radical type, which was not observed in this study.

However, this g-factor is notably higher than those reported in our previous study (Runberg et al., 2020), which were 2.0028 $\pm$ 0.0001. The soot for both studies was obtained in an identical manner, but the current study appears to have a much higher concentration of oxygen- or nitrogen-centered radicals. This is likely due to the use of a diffusion flame, which cannot control



the air-to-fuel ratio, and oxygen or nitrogen were likely incorporated during combustion. While it is outside the scope of this study, it would be useful to control for this factor in future investigations.

A second parameter that can be used to characterize an EPFR signal is its peak width ($\Delta B_{p-p}$), measured as the distance along the x-axis (magnetic field, B) between the maximum and minimum absorbances (peaks) of the derivative EPR signal. Both the peak width and the line shape (discussed below) provide information about how the varying spin environments in a complex sample interact with one another. Changes to $\Delta B_{p-p}$ indicate changes to the composition of the bulk sample (Sorin and Vlasova, 1973).

There was no difference in peak width between any of the dry samples (fresh, photo-aged, and dark-aged), indicating no impact to the spin environments from the photo-aging of dry soot. However, there were small but statistically significant differences between the dry and the wetted samples (t-tests done at 95% certainty, with $p < 0.05$) (Figure 3). The $\Delta B_{p-p}$ decreased for both aged soot samples (dark and light) upon submersion and agitation in water. For soot that had been photo-aged dry, $\Delta B_{p-p}$ decreased 4.4% in the dark and 5.7% when exposed to light. For soot that had been dark-aged dry, $\Delta B_{p-p}$ decreased 5.2% in the

dark and 5.8% when exposed to light. Additionally, when comparing the peak width of the wetted soot samples, soot that had been kept in the dark for dry-aging had $\Delta B_{p-p}$ 2.3% smaller than soot that had been dry-aged in light.



**Figure 3 Box plot of peak-to-peak widths ($\Delta B_{p-p}$) for each of the soot stages. Error bars are the standard deviation of three trials. The average is shown as a white triangle. Samples A, B, and C were dry soot, samples D-G had been wetted and then dried prior to analysis.**

Lineshape is another way to characterize an EPFR signal. EPR lines are complex functions comprised of the signals from each unpaired electron in the bulk sample. The bulk signal is modelled as a summation of each individual signal, using Gaussian and Lorentzian line functions. The more homogenous the sample, the more Lorentzian in character the lineshape will be. A more complex sample will result in more hyperfine coupling, which results in an increase in Gaussian character (Petrakis, 1967).

In this study, EPR lineshapes were seen to change in the wet portion of the aging process, with a significant decrease in Lorentzian character and an increase in Gaussian character (Figure 4) indicating a decrease in the homogeneity of the EPFR





environments in the soot and an increase in hyperfine interactions. Gaussian contribution increased from an average of (0.06 ± 0.03) mT across stages I and II to (0.22 ± 0.01) mT in stage III, an increase of 133%. Lorentzian contribution decreased 17%

from a combined average of (0.51 ± 0.02) mT for dry soot to (0.40 ± 0.01) mT wetted soot. This is an indication that the variability of electron environments within the sample increased with exposure to the water, but not to the light. This is counter to our previous study which did not result in any lineshape changes due to exposure to water. However, in that study, the soot was submerged and not agitated. So, the lineshape change seen here is likely due to the continuous agitation that occurred during the photoreaction, which broke up large soot fragments. An increase in soot density was observed between dry soot,

and soot that had been wetted (see Table A1), supporting this assessment. Soot density was determined by dividing the mass of the soot by the height of the soot in the capillary tube.



**Figure 4 Changes in Gaussian and Lorentzian contributions to the EPR lineshape. Error bars are the standard deviation of three trials. The average is shown as a white triangle. Samples A, B, and C were dry soot, samples D-G had been wetted and then dried**

**prior to analysis.**



## 3.2 Hydroxyl radical (OH) Formation

The ability of soot to produce OH was measured in aqueous conditions to mimic the conditions in a droplet of cloud water. Analysis was separated by soot that had been previously photo-aged and soot that had been previously aged in the dark. No measurable OH was observed in either of the dark reactions (Figure 1, samples D and F), confirming that light is required for

OH production. For soot slurries that were exposed to light, soot that had been photo-aged for 24 hours previously had the lowest OH concentrations at $0.38 \pm 0.02$ µM. This is not significantly different ($p = 0.06$) than controls done in ultrapure water, which were $(0.52 \pm 0.06)$ µM. Soot that had been dark-aged for 24 hours prior to the photoreaction in water resulted in significantly higher concentrations of OH with $(0.89 \pm 0.05)$ µM (Figure 5) ($p < 0.005$). This is consistent with a recent study reporting a decrease in OH concentrations in wildfire smoke plumes downfield from the point source, vs the same plume

measured nearer the source (Akherati et al., 2022). A laboratory study showed reduced ROS activity in aged PM as well (Gehling et al., 2014). In that study, particulate matter was aged at room temperature in phosphate buffered saline solution to maintain a physiologically neutral pH, but no application of light was executed. The authors reported a decrease OH formation of about 11% after one day.



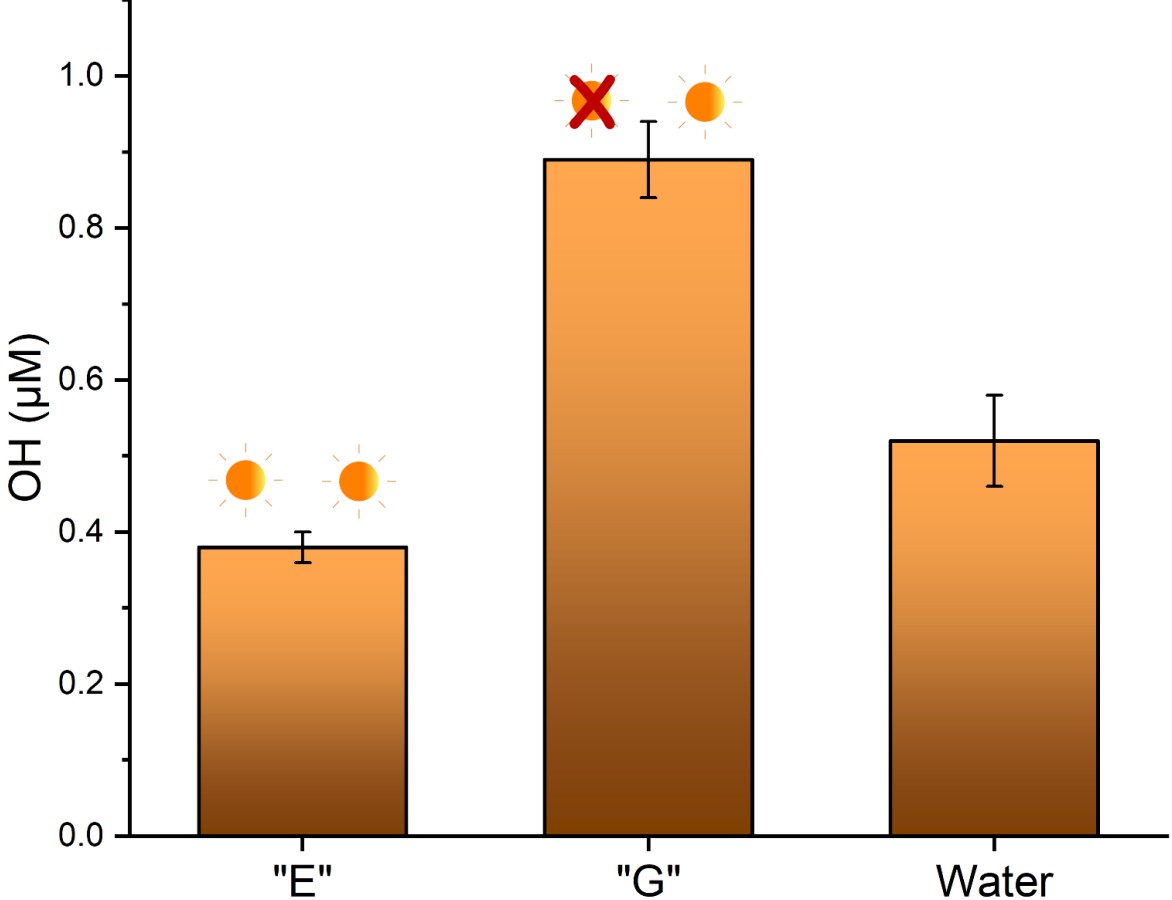

**Figure 5 Concentrations of OH measured from lighted reactions of soot that had been previously aged for 24 hours in the light (E), aged for 24 hours in the dark (G) and an ultrapure water control.**

### 3.3 GC-MS

GC-MS analysis was done to qualitatively determine if there was a change in the number of organic-soluble compounds in the soot at different stages of the aging process. In all dry soot samples (samples A, B, and C in Figure 1), only one compound

was seen, at a retention time of 9.13 min. (Figure 6, top row). Analysis of the fragmentation pattern of this peak indicated no obvious nitrogen- or oxygen-containing fragments. For all samples that had been submerged in water (Samples D – G in Figure 1), there were several new peaks observed (Figure 6, middle and bottom rows). Mass spectral comparisons to the NIST MS database did not definitively identify any of the compounds, but all display a distinctly hydrocarbon-like fragmentation pattern (Figure A2).



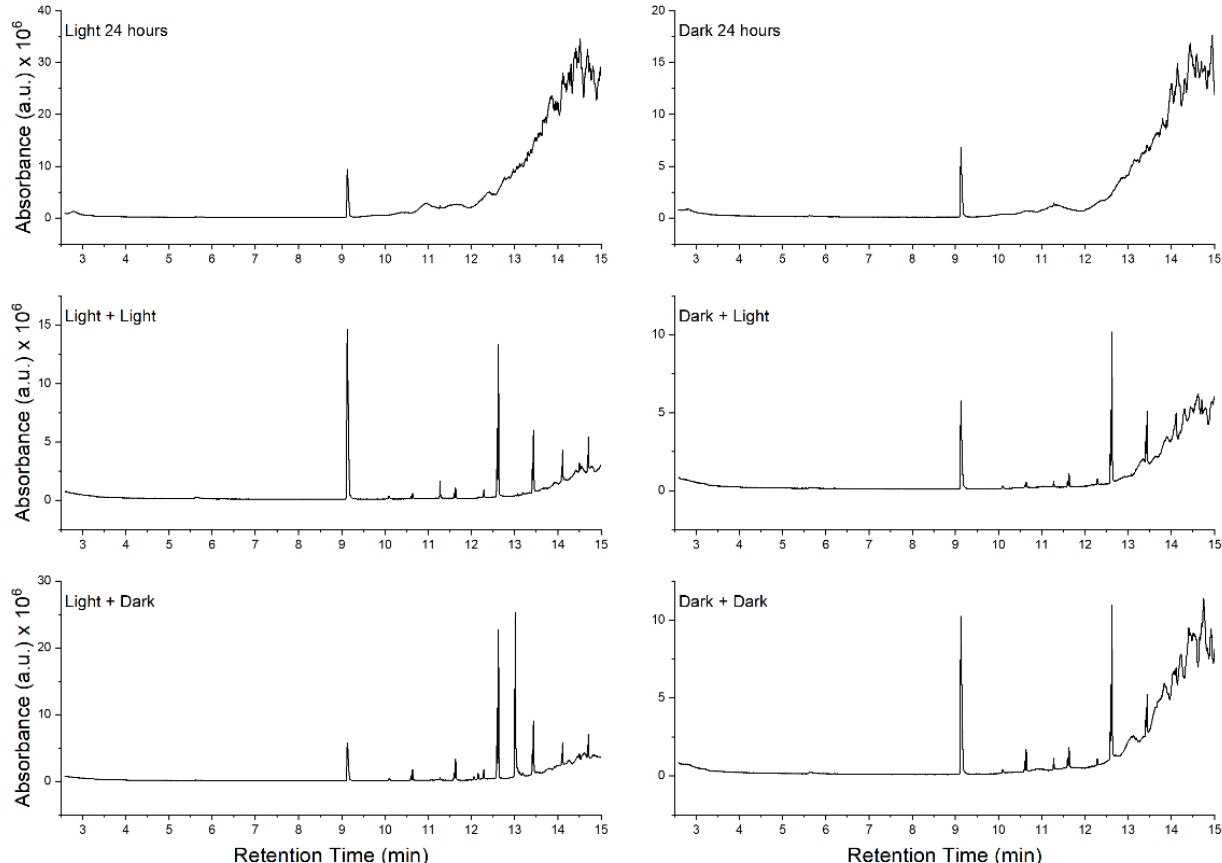

**Figure 6 Gas chromatograms of soot at each step of the aging process. The left column contains chromatograms of soot extracts taken from the photo-aged pathway (samples B → E & D), and the right column contains extracts of soot that followed the dark-aged pathway (samples C → G & F). The first row is extracts from the soot after initial 24 hours of aging. The second row is soot that was photo-aged in water for 16 hours, and the bottom row is soot that was dark aged in water for 16 hours.**

There is indication that oxygen is being incorporated into the new peaks observed in samples that had been wetted. For example, analysis of the fragmentation pattern on the peak which eluted at 13.44 min. suggests a McLafferty rearrangement (Burgers and Terlouw, 2016), resulting in an acylium cation with m/z = 43 and a McLafferty fragment at m/z = 57, implying the presence of at least one carbonyl group (Figure A3).

**3.4 Fourier transform infrared (FTIR) spectroscopy**

Fourier transform infrared (FTIR) spectroscopy was done for all soot samples to assess bulk changes in functional groups. All dry soot samples appear nearly identical, indicating no changes to functionality via dry aging. There is new peak formation in all samples that had been wetted, indicating that the change is likely due to the prolonged agitation in water rather than due to photoreactions of the soot or with any OH formed during the reaction. New peaks were observed at about 2915 cm$^{-1}$ and 2850 cm$^{-1}$, consistent with an increase in C—H stretch, and at 1207 cm$^{-1}$ and 1140 cm$^{-1}$, which indicate a C—O stretch (Figure 7).



Additionally, several peaks are observed in all samples: a peak at 1575 cm$^{-1}$ suggests the presence of N—O stretching, a peak at 1730 cm$^{-1}$ may be C=N, and a peak at 2090 cm$^{-1}$ may be C=C=N, offering evidence of the nitro-components suggested by the g-factor of this soot, and was likely introduced during the combustion process.

**Figure 7 FTIR spectra of all soot samples. Fresh soot (A), dry soot after 24 hours of illumination (B), dry soot after 24 hours in the**
**dark (C), soot that was photo-aged for 24 hours, followed by 16 hours in the dark, in water (D), soot that was photo-aged for 24 hours, followed by 16 hours in the light, in water €, soot that was dark-aged for 24 hours, followed by 16 hours in the dark, in water (F), soot that was dark-aged for 24 hours, followed by 16 hours in the light in water (G). Red circles indicate new peaks associated with incorporation of oxygen.**





## 4 Conclusions

EPFR characteristics of combustion particles, such as g-factor, line shape, peak width, and spin concentration, are unaffected by photoaging, however, soot's ability to generate OH when exposed to sunlight diminishes drastically after light exposure. This implies several possibilities: First, it may be something other than EPFR within the particles that drives the OH formation, or, second, that only EPFR on the surface of the particles are available for OH formation, while the bulk of the EPFR component is bound within the particles where they are not impacted by the light. Combustion particles that were not photoaged

produced almost 60% more OH than photoaged soot did. It is possible that any changes to the surface EPFR due to exposure to light was muted by the larger concentration of EPFR found within the particles, resulting in no meaningful change to the EPFR concentration or characteristics of the exposed soot. The lack of changes to chemical composition of the soot upon irradiation, as seen in the GC-MS and UV-Vis analysis, support the hypothesis that only EPFR on the surface of the particle are available for OH formation.

Other environmental factors - specifically the chemical composition of natural cloud waters - may impact the way that combustion particles age in the environment, including some properties of EPFR, and future work should include this variable. Natural cloud waters are a complex matrix containing compounds such as sulfates, nitrates, metals, and organic matter. Previous work has indicated that natural cloud water has a much lower pH than pure water, and the lower pH of natural cloud water has been implicated in increased solubility of organic aerosols (Li et al., 2020), which may influence photo induced OH

formation from soot.

       In summary, EPFR characteristics of soot do not change upon photoaging, but the soot's ability to generate OH is greatly reduced.  This has implications to human health, because it implies that freshly generated soot, as from a wildfire, is potentially more hazardous upon inhalation than is soot that has travelled further away from its point source. This could result in an increase in negative health outcomes for people living close to frequent wildfires. Due to global climate change, the frequency

and intensity of wildfires is expected to increase, which means that more people will become exposed to freshly generated soot particles.

## Appendix A

**Table A1 EPR data for each soot sample collected. Samples were collected in triplicate from each batch.**

| Sample | Q Value | Freq (GHz) | g factor | Gaussian (mT) | Lorentzian (mT) | Soot density (mg mm$^{-1}$) | spins mg$^{-1}$ | |
|---|---|---|---|---|---|---|---|---|
| Fresh soot 1 | 4105 | 9.659 | 2.0057 | 0.0817 | 0.5066 | 0.052 | 3.24E+16 | ± 1.26E+14 |
| Fresh soot 2 | 4104 | 9.658 | 2.0055 | 0.0668 | 0.5046 | 0.056 | 3.50E+16 | ± 3.48E+14 |
| Fresh soot 3 | 4105 | 9.659 | 2.0056 | 0.0099 | 0.5244 | 0.104 | 1.95E+16 | ± 1.58E+15 |
| Dry photo-aged soot 1 | 4105 | 9.659 | 2.0055 | 0.0523 | 0.5232 | 0.076 | 2.93E+16 | ± 2.00E+14 |



| | | | | | | | | |
|---|---|---|---|---|---|---|---|---|
| Dry photo-aged soot 2 | 3518 | 9.658 | 2.0056 | 0.0907 | 0.4908 | 0.086 | 3.16E+16 | ± 2.65E+14 |
| Dry photo-aged soot 3 | 4105 | 9.659 | 2.0056 | 0.0299 | 0.5216 | 0.081 | 2.85E+16 | ± 4.16E+15 |
| Dry dark-aged soot 1 | 4104 | 9.659 | 2.0057 | 0.0818 | 0.4877 | 0.108 | 2.57E+16 | ± 2.89E+14 |
| Dry dark-aged soot 2 | 4104 | 9.657 | 2.0057 | 0.0808 | 0.4867 | 0.100 | 3.11E+16 | ± 1.52E+16 |
| Dry dark-aged soot 3 | 3518 | 9.659 | 2.0056 | 0.0548 | 0.5242 | 0.067 | 4.36E+16 | ± 1.72E+15 |
| Dark-aged soot, dark OH 1 | 4105 | 9.660 | 2.0056 | 0.2084 | 0.4089 | 0.161 | 3.13E+16 | ± 7.81E+14 |
| Dark-aged soot, dark OH 2 | 4105 | 9.660 | 2.0055 | 0.2123 | 0.4059 | 0.180 | 2.87E+16 | ± 2.08E+14 |
| Dark-aged soot, dark OH 3 | 3519 | 9.660 | 2.0056 | 0.2100 | 0.4039 | 0.178 | 2.82E+16 | ± 1.13E+15 |
| Dark-aged soot, light OH 1 | 4105 | 9.659 | 2.0056 | 0.2152 | 0.3934 | 0.200 | 2.42E+16 | ± 2.00E+14 |
| Dark-aged soot, light OH 2 | 4105 | 9.660 | 2.0056 | 0.2101 | 0.4062 | 0.167 | 3.37E+16 | ± 1.15E+15 |
| Dark-aged soot, light OH 3 | 4105 | 9.660 | 2.0056 | 0.2078 | 0.4073 | 0.180 | 3.03E+16 | ± 1.65E+15 |
| Photo-aged soot, dark OH 1 | 4105 | 9.659 | 2.0055 | 0.2393 | 0.3754 | 0.220 | 2.58E+16 | ± 7.21E+14 |
| Photo-aged soot, dark OH 2 | 3518 | 9.659 | 2.0056 | 0.2266 | 0.3809 | 0.236 | 2.29E+16 | ± 5.96E+15 |
| Photo-aged soot, dark OH 3 | 4105 | 9.659 | 2.0056 | 0.2262 | 0.3813 | 0.244 | 2.72E+16 | ± 1.08E+15 |
| Photo-aged soot, light OH 1 | 3519 | 9.661 | 2.0056 | 0.2238 | 0.3790 | 0.193 | 2.93E+16 | ± 5.48E+15 |
| Photo-aged soot, light OH 2 | 3518 | 9.660 | 2.0056 | 0.2288 | 0.3792 | 0.207 | 2.92E+16 | ± 3.46E+14 |
| Photo-aged soot, light OH 3 | 3518 | 9.658 | 2.0056 | 0.2274 | 0.3794 | 0.227 | 2.61E+16 | ± 7.64E+15 |






**Figure A1 Representative EPR spectra of three soot samples at different stages of photo-aging, Fresh (black, solid line), soot aged for 24 hours under simulated sunlight (red dash-dot), and soot photo-aged twice, once dry, and once wet (blue, dash-dash). Signals have been baseline corrected and intensity normalized.**




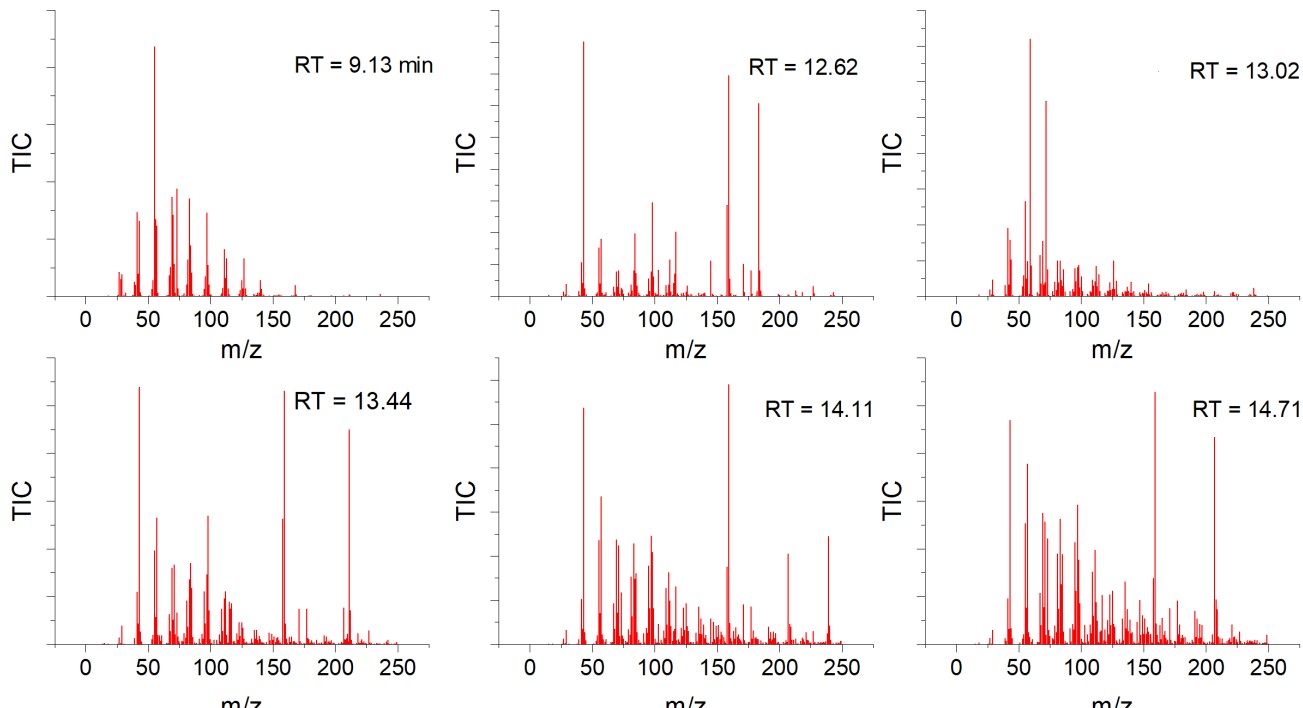

**Figure A2 Mass Spectra of select peaks from the GC obtained from "Light->Dark" soot extracted into acetonitrile. Retention times refer to those seen in Figure 6.**



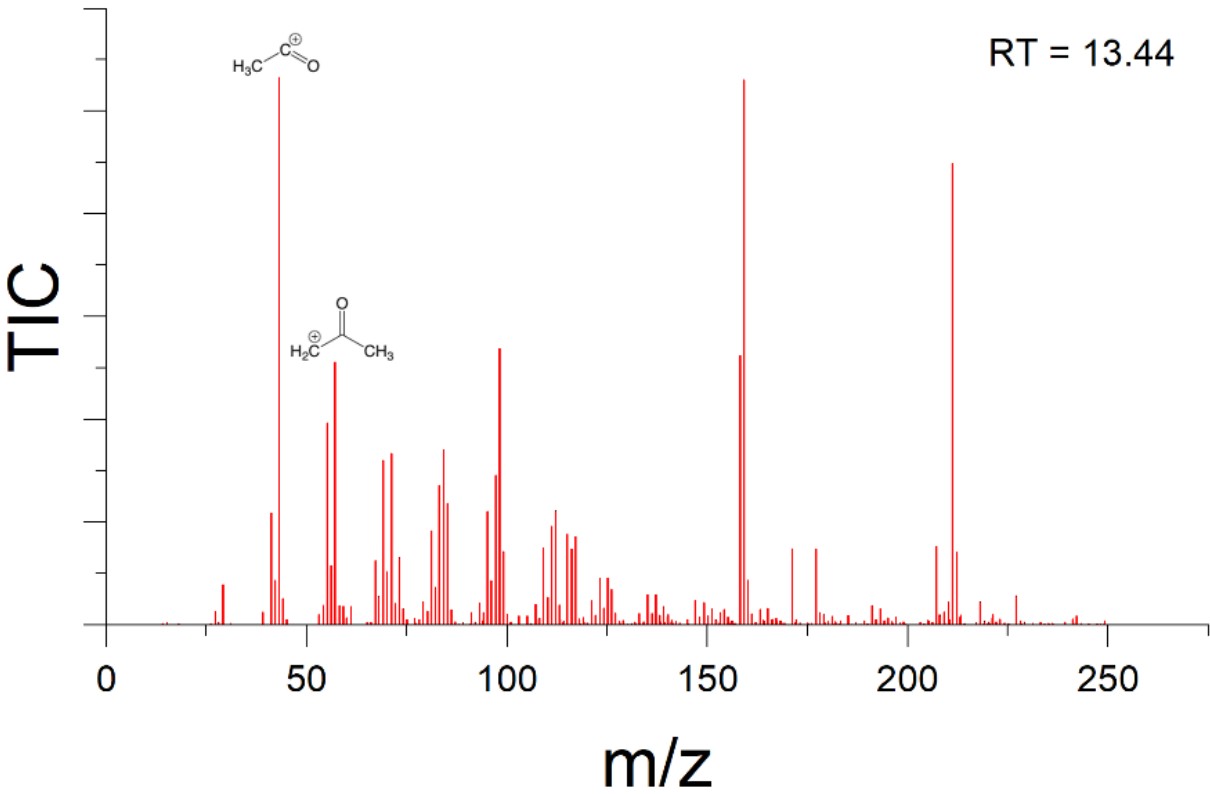


**Figure A3 Mass spectrum of the peak seen at retention time 13.44 minutes.**

**Data Availability**

Data described in this manuscript can be accessed at repository under data doi
(https://doi.org/10.56902/Chem.Data.2023.1)(Majestic and Runberg, 2023)

**Author Contribution**

Heather L. Runberg: Methodology, Software, Validation, Investigation, Writing - Original Draft, Data Curation, Visualization

Brian J. Majestic: Conceptualization, Resources, Writing - Review & Editing, Supervision, Project administration

**Competing interests**

The authors declare that they have no conflict of interest.



**Acknowledgements**

The authors thank Professors Sandra and Gareth Eaton at the University of Denver for allowing us access to their lab's EMX nano for this research, as well as for their expertise and assistance in analyzing EPR spectra.

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
