# Peer review of "Measurement Report: Atmospheric aging of combustion-derived particles: impact on stable free radical concentration and its ability to produce reactive oxygen species in aqueous media"

_EGUsphere, 2023_

## Author Comment (AC1)

EPFRs character change and ability to induce OH formation under influence of light and water were studied to simulate the environmental fate of EPFRs in this research. EPFRs were characterized fully by many instrumental or fitting methods. The results are valuable. In general, after minor revision, the manuscript can be considered for publication on EGUsphere.

1. Have you verified that materials you used could not interference the results during the experiment? Transition metals were reported to influence formation and stabilization of EPFRs. The disposable aluminum baking dish and aluminum foil used in this research might have influences. Please clarify that.

Response:

The aluminum baking dish was not in contact with the soot until after it had cooled, long after the formation period, so there is not likely to be an impact from it on the formation of combustion derived EPFRs.

Additionally, while we agree that there is a transition metal-mediated pathway toward formation of EPFR, there is also a pathway which involves the growth of organic matrices which stabilize the radicals formed within (Liao et al., 2014). Because of the fuel-source and controlled environment, the second pathway is governing here.

Lines 52-54 has been modified as follows:

*"The soot was collected from the funnel into a disposable aluminum baking dish (Reynolds KITCHENS® mini loaf pan) as soon as it was cool . The soot was transferred to the aluminum baking dishes only after it was fully cooled to prevent any interference from the foil on the EPFR structure."*

2. Section 2.2, you mentioned there was a storage period of the EPFR samples. Considering the inevitable decay and high reactivity of free electron (even though in EPFRs), please provide the storage time of each sample and try to give evidence for not change of EPFRs during the storage.

Response:

EPFR samples were stored in a standard freezer from 2 to 7 days. EPR spectra are not statistically different for any of the samples, indicating that the aging of the EPFRs in the soot while under storage was insignificant. Additionally, we have a previous publication that shows very little change in any EPFR parameters of soot generated in the same manner after 30 days storage at room temperature (Runberg et al., 2020).

Lines 70-71 have been modified as follows:

*"All samples were stored in a freezer when not being analyzed, with storage times ranging from two to seven days (**Error! Reference source not found.**)."*

Additionally, Table A1 has been updated to include the storage times for each sample.

3. Line 134, though the g value in samples A-G showed no obvious change, it still could not conclude whether there was no radical type change. There was no hyperfine splitting information in the EPR spectra (Figure A1), making it difficult to deduce the structure of EPFRs. In addition, except for the influence of temperature, there were probably more than one kind of radical in each of the samples to give a single, broad peak. The sentence "Changes in g-factor indicate a change in radical type, which was not observed in this study." was suggested to be modified.

Response:

We thank the Reviewer for this insightful comment. The composition of persistent radicals found in soot is a complex matrix of carbon and oxygen centered radicals. The EPR signal is, as the reviewer suggests, a superposition of the signals from all radical types within this matrix. This results in a signal with little to no hyperfine splitting.

Lines 136-139 have been modified as follows:

*"Because of the lack of hyperfine splitting in the EPR signal, it can be deduced that the soot is comprised of a large number of organic radicals which cannot be uniquely defined. Changes in g-factor indicate a change in radical type, for example, a shift from predominantly carbon-centered to predominantly oxygen-centered radicals, which was not observed in this study."*

4. Line 145, please add discussion about possible composition change of samples based on the different $\Delta B_{p-p}$ data.

Response:

The small decrease in $\Delta B_{p-p}$ observed between dry and wetted soot samples is likely an artifact of the increase in Gaussian composition of the signal. Gaussian curves have a narrower profile than do Lorentzian curves with the same intensity (Sorin and Vlasova, 1973).

To clarify this to the reader, Lines 155-157 have been modified as follows:

*"This decrease in $\Delta B_{p-p}$ is likely an artifact of the increase in Gaussian character of the lineshape for samples that had been submerged and agitated in water. This is discussed further in the next section."*

5. Could you give possible EPFR structures based on the characterization data? And if possible, try to give the formation mechanism.

Response:

We thank the Reviewer for this suggestion. As there are many structures of EPFRs in soot, the mechanisms of EPFR formation are outside the scope of this study. More controlled

experiments are required to discern mechanisms. We have a parallel manuscript in review in the Journal of Physical Chemistry that further explores a more detailed formation mechanism. The focus here is the impact of environmental factors on changes to EPFR after formation occurs.

---

## Author Comment (AC2)

This manuscript investigated the impacts of atmospheric aging of combustion-derived particles on the EPFRs concentration and its ability to produce ROS in aqueous media. This issue and findings are significant, the experimental design and measurements are also ingenious. Some comments of improvements are suggested for considering as follows:

General comments:

We thank the Reviewer for the kind comments.

Revisions are needed to present results and support discussions appropriately. There are three main points worthy to notice.

1. The conclusion and significance of this study should be carefully summed up. For examples:

Thank you for the comment. To better summarize, the manuscript has been modified as noted below.

First, the author emphasized the importance of soot in PM2.5 (Line 41), but there was no particle size analysis for samples in this study.

Response:

The Reviewer is correct that particle size was not addressed here. This was a preliminary study to measure impacts of sunlight on EPFR concentrations and composition in bulk soot. However, in conditions similar to ours (heptane instead of hexane), the particle size of soot is generally in the ultrafine (< 100 nm) range (Xiong et al., 2018). Thus, comparing our results to PM2.5 is still valid.

Second, the soot generated from hexane in this study as model particle is not the main byproducts of wildfire (Line 245). It's more likely from the fuel combustion, as the hexane is widely used in the fuel and industry.

Response:

We agree that this study more closely resembles fuel exhaust (i.e. internal combustion engine) emissions than forest fires, however, this is a preliminary study to measure impacts of sunlight on EPFR concentrations and composition in bulk soot. Hexane was chosen as a fuel because it is

homogenous when compared to biofuel sources, such as those in a wildfire. Future studies should be done that use different fuel sources and explore more variables.

The following has been added to line 256-257 to emphasize this:

*"Future studies should investigate the differences in EPFR characteristics and OH production using multiple fuel sources, including biofuels such as seen in wildfires."*

Third, the EPFRs in soot from real combustion process was verified to be EPFRs-metal conjugate, which might pose potentially more health effects.

Response:

We agree that the metals-EPFR may complicate and/or exacerbate the health effects of EPFR. In trying to understand each variable for this relatively newly discovered type of pollutant, we have focused on the non-metals in this paper. Our lab has parallel analyses underway where various relevant metals are being added to the fuel, however this study was not focused on EPFR formed from metal-mediated processes, and the health impacts from those is outside the scope of this study.

2. The materials and methods section seems uninformative to readers. Since many techniques have been used in this study, detailed and clearly descriptions of these methods are strongly recommended to support the coming results. For examples:

Line 60: How was the "final soot concentration was 45 ppm (m/v) " prepared? Any micro-balance used for soot weight quantification should be told?

Response:

 Yes, a microbalance was used to mass the soot. To clarify this, lines 61-64 has been modified as follows:

*"Using a microbalance, 0.0225 ± 0.0005 g of soot was combined with 500 mL of MQ water for a  final soot concentration  of 45 ± 0.5 ppm (m/v) for slurries that were used for OH measurements This is in line with concentrations of PM$_{2.5}$ obtained in previous work (Leresche et al., 2021) For slurries used for all other analyses 118 ± 2 ppm (m/v) for  was used."*

Line 89: Why this concentration of SBA used in this study? How to determine the concentration of SBA used? Were there any preliminary experiments for supports?

Response: 1 mM SBA was used because it is in great excess of what was expected of the OH radicals produced (>1.0 μM). This ensured that SBA was readily available to react with any OH formed during the reactions. This method is discussed in detail in (Runberg and Majestic, 2022)

To clarify this, the line 92-94 of the text has been modified as follows:

*"SBA (1.0 mM) was added to 45 ppm soot slurries just prior to beginning the photoreaction. This concentration was selected to be in substantial excess of the expected OH concentration, allowing the probe to capture the majority of the OH formed during the reaction."*

Line 90: Why 24 h for soot aged but 16 h for photoreactions here?

In previous studies in our lab (Haynes et al., 2019; Runberg and Majestic, 2022), the photoreaction period for OH formation was 16 hours to mimic the longest amount of sunlight possible near the equator. This was continued with this study to be able to compare the results to the previous study. A 24-hour period for the dry aging was used to maximize measurable differences between photo-aged and dark-aged soot.

> **Commented [BM1]:** Not sure I understand this one…
>
> **Commented [HR2R1]:** I'm not sure how to explain it. I just wanted to make sure the dry/fresh soot had an opportunity to undergo any photochemical reactions it might undergo before starting the reactions in the water.

3. The results and discussion of OH production and measurement are not solid and convincible. This part should be paid cautious consideration.

Line 181: How were the controls done? It seems that there was no OH produced by soot slurries exposed to light and photo-aged previously. The results could come from SBA photo degradation, not only from the reaction with OH which was produced by soot samples.

Response: Controls of SBA in water (no soot) were done using the same method used for soot slurries, in both lighted and dark reactions, to determine the concentration of p-HBA formed from SBA in water alone. These results are reported in the manuscript (lines 181-182). We are unaware of any pathways for the formation of p-HBA from SBA that do not involve OH, either as a radical or as a hydroxide. Additionally, p-HBA is a synthesis product, not a degradation one.

Line 183-184: "This is consistent with a recent study reporting a decrease in OH concentrations in wildfire smoke plumes downfield from the point source, vs the same plume measured nearer the source (Akherati et al., 2022)". But soot with dark-aged and then exposed to light can produce higher OH.

Response:

The Reviewer is correct; as the smoke plumes travel away from the point source, they are photoaged, and therefore result in lower concentrations of OH further from the source. Closer to the source, where the smoke has not yet been aged, more OH is produced.

This has been clarified in the text, lines 193-194, as follows:

*"This is consistent with a recent study reporting a decrease in OH concentrations in wildfire smoke plumes downfield from the point source, vs the same plume measured nearer the source (Akherati et al., 2022). This indicates that smoke which has been exposed to sunlight for a longer period of time (i.e. the downfield plume) results in lower concentrations of OH."*

Line 185-188: That study (Gehling, 2014) measured the OH from dark-aged EPFRs containing-PM suspension, which is different from current study.

Response:

This is an interesting observation.  Gehling likely observed this due to the phosphate buffered saline solution used.  Lines 197-198 have been added to clarify this.

*"A laboratory study showed reduced ROS activity in aged PM as well (Gehling et al., 2014). In that study, particulate matter was aged at room temperature in phosphate buffered saline solution to maintain a physiologically neutral pH, but no application of light was executed. The authors reported a decrease OH formation of about 11% after one day. This implies that other factors may play a role in the aging of soot, in addition to sunlight."*

> **Commented [BM3]:** Can you be more specific?  What type of fluid did Gehling use?  And, should be put anything relating to this in the manuscript?

Line 228: "only EPFR on the surface of the particles are available for OH formation", but the samples were well mixed and stirred in this study.

Response:

Yes, they wereYes, only EPFR on the surface were available for reaction. EBut even in a well stirred slurry (as in this study), the interior of the particles is not accessible by the water or light. Although well-mixed, Nnote that this is still a suspension and not a solution.

Line 233-234: "The lack of changes to chemical composition of the soot upon irradiation, as seen in the GC-MS and UV-Vis analysis, support the hypothesis that only EPFR on the surface of the particle are available for OH formation". Doubt. The OH formation in this study seems more affected by the light.

Response:

The Reviewer is correct that the OH formation was strongly affected by light.  This supports the fact that this is likely a surface-mediated process, as the light does not penetrate the core of the particle.  Additionally, the fact that the overall bulk composition did not change supports that only the surface was affected.

**References**

Haynes, J.P., Miller, K.E., Majestic, B.J., 2019. Investigation into Photoinduced Auto-Oxidation of Polycyclic Aromatic Hydrocarbons Resulting in Brown Carbon Production. Environmental Science and Technology 53, 682–691. https://doi.org/10.1021/acs.est.8b05704

Runberg, H.L., Majestic, B.J., 2022. Hydroxyl radical (OH) formation during the photooxidation of anthracene and its oxidized derivatives. Atmospheric Environment 286, 119214. https://doi.org/10.1016/J.ATMOSENV.2022.119214

Xiong, G., Boot, M., Zhou, L., Reijnders, J., De Goey, P., 2018. Soot Particle Size Distribution Measurements in Laminar Diffusion Flames of *n*-Heptane with Oxygenated Aromatic Fuel Additives by Time-Resolved Laser-Induced Incandescence. Energy Fuels 32, 11511–11518. https://doi.org/10.1021/acs.energyfuels.8b02012

---

## Author Response (AR2)

*Comment 1*
*[1] I disagree with the use of "OH concentrations" in the manuscript (e.g. l. 93, l. 189 etc.). The authors do not measure in situ concentrations of OH (which are much lower at any given point in time), but rather use a probe to estimate time-integrated OH production. I suggest to replace "OH concentration" with "OH production" were applicable.*

Response:

We thank the Editor for the suggestion. References to "OH concentration" have been modified as noted below.

Figure 1, line 77
**"OH  production  was measured for stage III."**

Lines 96-97
"Each sample was run in triplicate and final p-HBA concentrations were averaged."

Lines 100-103
"Total  OH  production during the course of the reaction  was determined by dividing the measured p-HBA concentration by the fraction of OH that forms p-HBA after 16 hours, previously determined to be 0.56 (Runberg and Majestic, 2022)."

Lines 187-188
"For soot slurries that were exposed to light, soot that had been photo-aged for 24 hours previously had the lowest OH  production at $0.38 \pm 0.02$ µM."

Lines 189-194
"Soot that had been dark-aged for 24 hours prior to the photoreaction in water resulted in significantly higher  OH production,  at $(0.89 \pm 0.05)$ µM (Figure 1) ($p <$ 0.005). This is consistent with a recent study reporting a decrease in OH  formation in wildfire smoke plumes downfield from the point source, vs the same plume measured nearer the source (Akherati et al., 2022). This indicates that smoke which has been exposed to sunlight for a longer period of time (i.e. the downfield plume) results in lower  OH production."

Figure 5, Lines 201-202
**"Figure 1  Production of OH measured  after 16-hour lighted reactions of soot that had been previously aged for 24 hours in the light (E), aged for 24 hours in the dark (G) and an ultrapure water control."**

*Comment 2*
*[2] I suggest to add more disclaimers to the comparison with the work by Akherati et al. (2022). The authors already formulate with care ("... is consistent with ..."), but the brevity of the statement might lead a reader to overinterpret the findings. It could be made clearer that (i) not all gas-phase OH in an air pollution plume comes from photochemistry of soot in aqueous environment, (ii) OH produced in this way might not even enter the gas phase, and (iii) OH concentrations in a biomass burning plume will likely decrease naturally over time due to dilution. Alternatively or in addition, the authors can generalize the statement more, e.g. that a reduced capability of OH production after a period of photo-irradiation is also seen in more complex systems.*

We thank the Editor for the insightful suggestion. The following text has been added at line 194-195 to address this.

"While photoaging of airborne soot particles in a much more complex system such as a smoke plume is likely not the only reason for the decline (e.g. there are multiple OH production pathways and the plume is diluted), the same pattern of diminished OH production is reported."